# Distribution and habitat of the painted tree rat *(Callistomys pictus)*: Evaluating areas for future surveys and conservation efforts

**Andrés David Sarmiento Sanchez**[�], **Gabriela Alves-Ferreira**[iD][☐], **Neander Marcel Heming**, **Gastón Andrés Fernandez Giné**[iD]*

Departamento de Ciências Biológicas, Laboratório de Ecologia Aplicada à Conservação, Universidade Estadual de Santa Cruz (UESC), Ilhéus, Bahia, Brazil

☐ These authors contributed equally to this work.
* gastongine10@gmail.com

**Data Availability Statement:** All relevant data are within the manuscript and its Supporting Information files.

## Abstract

Knowledge of the potential distribution and locations of poorly known threatened species is crucial for guiding conservation strategies and new field surveys. The painted tree-rat (*Callistomys pictus*) is a monospecific, rare, and endangered echimyid rodent endemic to the southern Bahia Atlantic Forest in Brazil. There have been no records of the species published in the last 20 years, and the region has experienced significant forest loss and degradation. According to the IUCN, only 13 specimens had been previously reported, with 12 found in the north of Ilhéus and adjacent municipalities, and one recorded approximately 200 km away from this region, suggesting that its distribution might be wider. We aimed to search for unpublished and more recent records of the *C. pictus*, by consulting the gray literature (including Environmental Impact Study (EIA) reports, Brazilian Red Lists, and management plans of protected areas), scientific collections, online databases, and mastozoologists working in the region. We estimated the species' potential distribution using Ecological Niche Modeling to identify regions, municipalities, and protected areas most likely to support this species, based on factors such as climate suitability and forest cover. We reported three new sightings of the species, including the first within a protected area. We estimated suitable climate conditions across 23,151 km², of which 9,225 km² has a high potential for harboring the species. The area between Itacaré and Valença needs more extensive survey efforts as it has high habitat suitability and only one record has been confirmed there so far. Meanwhile, the region between Una and Ilhéus urgently requires habitat conservation initiatives. While the species may have a broader distribution than previously thought, its known occurrences are limited to a few locations, and suitable habitats are underrepresented in protected areas. Additionally, the rarity of sightings continues to indicate a concerning conservation status.

**Funding:** ADSS received a fellowship from Fundação de Amparo à Pesquisa do Estado da Bahia (FAPESB) during the preparation of this study. https://www.fapesb.ba.gov.br/ GAF received a doctoral scholarship from Coordenação de Aperfeiçoamento de Pessoal de Nível Superior during the preparation of this study (code 001). https://www.gov.br/capes/pt-br The funders had no role in study design, data collection and analysis, decision to publish, or preparation of the manuscript.

**Competing interests:** The authors have declared that no competing interests exist.

## Introduction

Species extinction is one of the most significant environmental issues caused by human activity and could lead to the disappearance of hundreds of species in the coming years [1–3]. Effective species conservation requires at least basic information about the geographic distribution, remaining habitat, and known locations of the target species [4,5]. Unfortunately, knowledge about these basic aspects is scarce for many species, especially those that are rare and cryptic, which constitute a significant portion of the endangered species in the tropics [6].

One way to minimize the deficit of knowledge about species distributions, known as the Wallacean shortfall [7], is to apply Ecological Niche Modeling (ENM) techniques. These techniques spatially predict areas with environmental conditions similar to those found in locations where a species is known to occur, indicating their potential distribution [8–10]. Particularly for species with small numbers of known occurrences and poorly understood distributions, although the limited data may lead to a potential loss in model accuracy [11,12], such predictions are valuable for identifying potential sites for targeted field surveys and for conducting preliminary assessments of the species' conservation status [12–17].

The painted tree-rat (*Callistomys pictus*, Pictet 1843) is a monospecific, rare arboreal echimyid rodent endemic to the southern Bahia Atlantic Forest in Brazil [18–20]. Their current geographical distribution is not well-documented [19]. The subfossil specimen discovered in Minas Gerais state [21] indicates a significant reduction in its geographic distribution since the Pleistocene [22]. According to the IUCN, only 13 specimens are known [19], of which the last was sighted in 2003 [18]. Most animals were observed in cocoa agroforests and native forests in a microregion of northern Ilhéus and two neighboring municipalities [18,22–26]. However, in 1997, this species was recorded for the first time outside this microregion, in the municipality of Elísio Medrado, located about 200 km north of Ilhéus [24], suggesting that its current distribution might be wider than previously thought [22].

The painted tree-rat is classified as "endangered" on the IUCN Red List [19] because its occupancy area is quite restricted (less than 500 km$^2$), its population is severely fragmented, existing in no more than five known locations (*sensu* IUCN [5]), and there is a continued decline in the quantity and quality of its habitat [19]. *C. pictus* exhibits the most extreme locomotor specializations for arboreality among echimyids [20,27], suggesting it is strictly arboreal and forest-dependent. The landscape in northern Ilhéus, where the painted tree-rat is predominantly found, mainly consists of shaded cocoa plantations and forest fragments, both described as the only habitat types used by this rodent species [22,23,25]. The loss and degradation of these habitats have recently intensified in this microregion due to the construction of a port complex (for the export of iron ore), access roads, strong urbanization, and policies that encourage logging and more intensive management of shaded cocoa plantations (Federal Law No. 12.651 and Decree No. 15.1808). Such local threats could severely impact the species' population, as it is found in only a few locations and has not yet been recorded in any protected areas [18,19].

In this context, three questions motivated us to carry out the present study: Have there been any new, unpublished locations where the species has been found? Where could the species potentially occur? Where should efforts for new surveys and conservation be focused? To address these interdependent questions, given the limited information on the current occurrence and distribution of the endangered painted tree-rat, we aimed to: (1) compile an occurrence database and search for new records of the species by consulting peer-reviewed literature, gray literature (material not peer-reviewed), scientific collections, online databases, and mastozoologists working in the region, providing the foundational data for subsequent analyses; (2) estimate the size of the species' potential occurrence area (climatically suitable

area), the total area of remaining habitat, and the extent of protected areas within this range, which is critical for both identifying key regions for the species' occurrence and evaluating its conservation status; and (3) identify regions, municipalities, and protected areas with the highest potential to harbor the species (based on climate suitability and forest cover), prioritizing areas for future surveys. Finally, to support the development of conservation policies for the target species, we provide recommendations for conservation actions based on the current knowledge of confirmed species occurrences and the regional representativeness of protected areas within its potential distribution.

## Methods

### Study area description

The study area is the Atlantic Forest domain [28,29]. We selected a portion of the biome distributed from the northern part of Rio de Janeiro state to the northern part of Sergipe, Brazil (37.58–45.93˚W; 11.5022.29˚S). This area (337,740 km$^2$) is larger than the species' range listed in the IUCN Red List (27,160 km$^2$; [19], Fig 1B). This discrepancy exists because a primary goal of the research is to identify potential new habitats for the species. This region encompasses two Atlantic Forest ecoregions, where the species has been reported (Fig 1B), each presenting distinct environmental conditions: Bahia Coastal Forests and Bahia Interior Forests are characterized by evergreen (ombrophilous) and seasonal (deciduous and semideciduous) forests, respectively [30]. The predominant climate in the region is tropical [31], with average annual temperatures ranging from 13.9˚C to 25.4˚C and average annual precipitation varying from 507 mm to 2444 mm [32]. Only 11.1% of the original forest remains in this region, and deforestation continues at a high rate [33].

### Occurrence data

To address the first objective, we compiled data on the occurrence of the painted tree-rat by reviewing literature (peer-reviewed articles) and gray literature (including Environmental Impact Study (EIA) reports, Brazilian Red Lists, and management plans of protected areas) [18,22,24,25,34,35], consulting ecologists and mastozoologists with regional expertise (M. Alvarez, C. Cassano, P.S. Gouveia, M. Freitas, K. Flesher, B.F. Rosa, G. Canale, R. Moura, D. Faria, R. Bovendorp, and F.C. Falcão), and searching information in scientific collections and online databases. We consulted curators at the following collections: CMARF–Coleção de Mamíferos Alexandre Rodrigues Ferreira, UESC, Ilhéus, Brazil; MZUFBA—Museu de Zoologia, UFBA, Salvador, Brazil; and Museum für Naturkunde, Berlin, Germany. Online databases included SpeciesLink(http://www.splink.org.br/), Global Biodiversity Information Facility (GBIF: http://www.gbif.org), Biodiversity Information Serving Our Nation (BISON: https://bison.usgs.gov), Vertnet(http://vertnet.org/), and Integrated Digitized Biocollections (iDigBio: https://www.idigbio.org). We did not consider data from interviews with local residents, fossils, or data without geographic coordinates. We verified the geographic accuracy of the data by comparing the provided coordinates with the descriptions of the collection sites (e.g., village, district, farm, and road stretch). We did not consider occurrence data derived from municipality and state centroids. When possible, we consulted the author of the record to ensure accuracy regarding the originally described location. In cases where data was published more than once, only the record with the most accurate geographic location was retained in the database. To reduce the sampling bias, we removed occurrences that were at least 1 km apart.

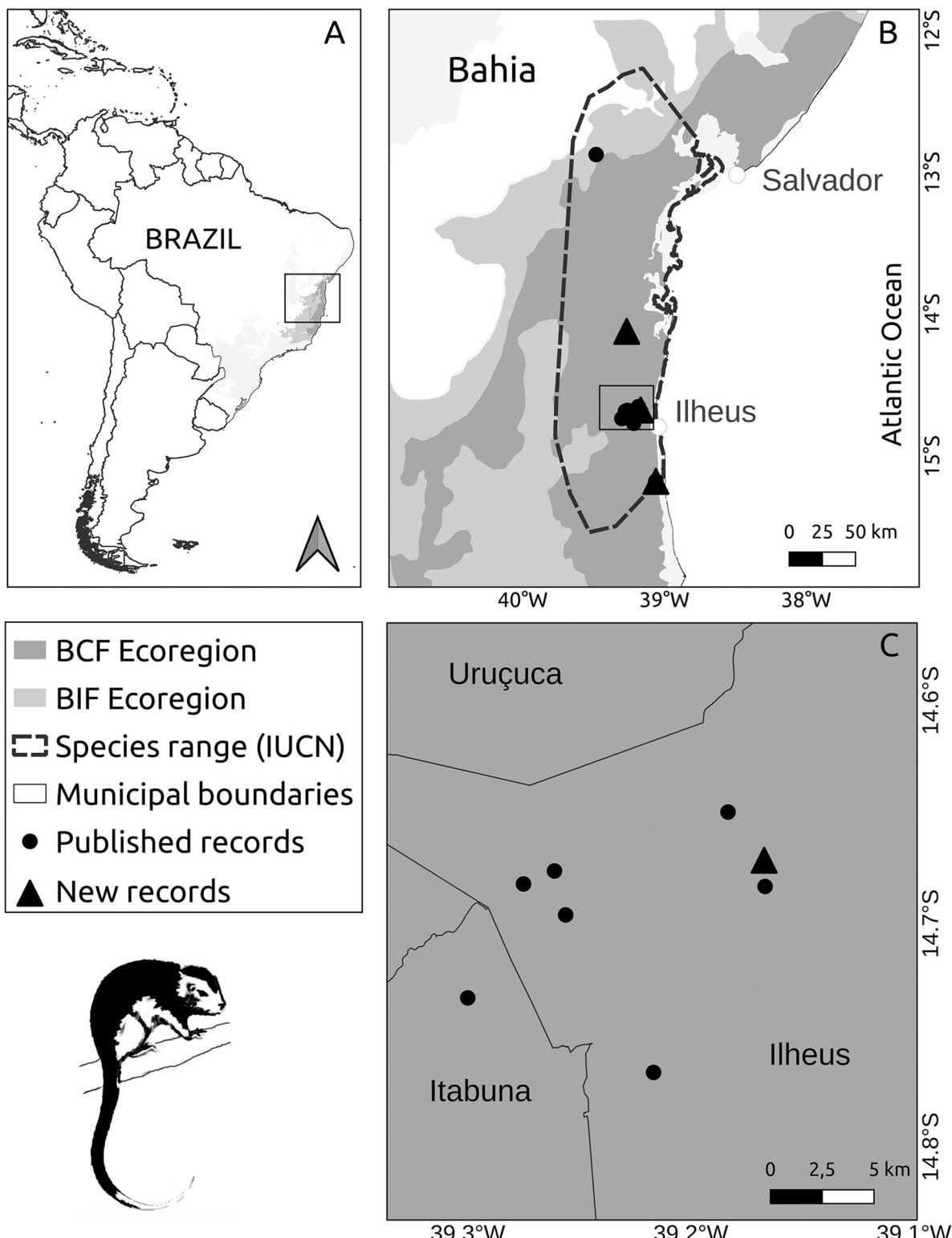

**Fig 1. Study area.** (A) Geographical area where the painted tree-rat (*Callistomys pictus*) was studied. (B) Map showing the polygonal distribution of the species (outlined by a dashed black line) as defined by the IUCN (Roach & Naylor 2019), the ecoregions of the Atlantic Forest (BIF: Bahia Interior Forests Ecoregion, and BCF: Bahia Coastal Forests Ecoregion), and the recorded locations of the painted tree-rat, categorized into historical records (black dots) and recent records post-2003 (black triangles). (C) Map showing the distribution of the painted tree-rat in the Northern Ilhéus microregion, where the species is most frequently observed.

## Environmental variables

We downloaded data for 19 bioclimatic variables for the years 1970–2000 from the WorldClim version 2.1 website, http://www.worldclim.org, with a spatial resolution of 30 arc-seconds (~1 km$^2$) [32]. After defining the calibration area (see below), we estimated Pearson's correlation coefficient and performed a cluster analysis to assess collinearity among the variables using the "select_vars" function of the *ENMwizard* R package [36]. We selected a set of bioclimatic variables with Pearson's correlation coefficients below 0.70 (S1 Fig). We selected six environmental variables to develop the models: Annual Mean Temperature (BIO1), Isothermality (BIO3), Temperature Seasonality (BIO4), Temperature Annual Range (BIO7), Precipitation of the Wettest Month (BIO13), and Precipitation Seasonality (BIO15).

## Ecological niche model

To address the second objective, we constructed the climatic niche models for the species using the occurrence data and bioclimatic variables. We used MaxEnt (version 3.4.4, [37]) because this algorithm typically performs well compared to others [38], even with a limited amount of occurrence data [39]. Using the "ENMevaluate" function of the *ENMeval* R package [40], we built and evaluated 135 models by combining nine regularization multiplier values (ranging from 0.5 to 4.5 in 0.5 increments) with all 15 possible combinations of the four feature classes (Linear = L, Quadratic = Q, Product = P, and Hinge = H). We defined the calibration area for model fitting by establishing a 1.5° (~155 km$^2$) buffer around a Minimum Convex Polygon (MCP) formed by all occurrence points of the target species [36,41,42]. We assumed this area is potentially accessible to the species and encompasses sufficient environmental heterogeneity to estimate the species' environmental or niche preferences.

To construct the models, 10,000 pseudo-absence points were randomly generated within the calibration area (background) using the default settings of MaxEnt. We estimated model evaluation metrics by partitioning the training and testing occurrence data using the n-1 jackknife method, recommended for databases with limited occurrence records [12,43]. Considering that data from 11 occurrences were used, the n-1 jackknife method involves using 10 occurrences to calibrate the model and one to evaluate it, repeating this process until all points have been used for testing [44,45]. We selected the "best" model based on its significance (partial ROC), lower omission rate (OR), and lower Akaike Information Criterion (AICc) values, following the methodology described in Cobos et al. (2019). The best model for the study area was generated using the "proj_mdl_b" function from the *ENMwizard* R package [36]. The output of the model in "Cloglog" format can be interpreted as continuous values indicating relative climatic suitability, ranging from 0 to 1. These values reflect the climatic similarity to the areas where the species is known to occur. Furthermore, we selected the top 10% of models using the same criteria and built an average ensemble model for comparison [46]. Although the results of the "best" single model and the ensemble model were similar (S2 Fig), we continued the analysis based solely on the "best" model because it provided slightly broader spatial predictions, aligning with our main objective to identify new potential areas for future surveys of the species.

## Potential area of occurrence and core area

To define the climatically suitable area for the target species, also referred to as the "potential occurrence area," we developed a binary model (0 or 1) from the continuous suitability values using a threshold defined by the minimum training presence (MTP). This threshold is set at the lowest suitability value found in areas where the species is known to occur. Consequently, the resulting binary model encompasses all known locations of the species.

To define the area of high suitability for the target species, referred to as the "core area," we developed a binary model using the 10th percentile training presence threshold (10ptp). This threshold delineates the suitable area by excluding the 10% most marginal locations of the species' known occurrences [47]. We assumed that the core areas are those with the greatest potential to harbor the species and where survey efforts should be primarily directed. We estimated the area (km$^2$) of the binary models generated by both thresholds (MTP and 10PTP) and assessed the climatic suitability gradient within these areas.

### Remaining habitat, municipalities, and protected areas

To address the third objective, we estimated the total area of remaining habitat and protected areas within the potential range and core area of the painted tree-rat. Additionally, we calculated the total area of remaining habitat within the protected areas and municipalities located in the species' potential and core occurrence areas. The municipalities and protected areas were ranked based on the total area of remaining habitat within each, to identify those with the greatest potential for supporting populations of the species and to guide future survey and conservation efforts.

For these estimates, we used the land use and vegetation map with a resolution of 30 m from Mapbiomas Collection 5 (https://mapbiomas.org/download), classifying the "natural forest formation" category as habitat. This category encompasses native forests and cocoa agroforests. These two vegetation types, collectively referred to as "forested habitats," are considered the sole habitats utilized by the target species [22,23,25,48]. To estimate the total area of remaining habitat within the municipalities and protected areas, we used the political boundaries of the municipalities [49] and the polygonal boundaries of the Brazilian federal, state, and municipal strictly protected areas (https://mapas.mma.gov.br/i3geo/datadownload), which correspond to IUCN categories Ia and II [50]. Given the low efficiency of management and conservation of non-strictly protected areas we decided to be more conservative by highlighting only those that have strict protection. All analyses were performed using R software [51].

## Results

### Occurrence records

In total, we compiled 23 occurrence records of paint-tree rats (S1 Table) including three new specimens sighted after 2003. In total, 19 specimens were deposited in scientific collections and 20 were previously reported in the peer-reviewed articles. Accurate geographical coordinates (excluding province, municipalities or district centroid) are known for only 11 specimens, which were used to build the ENMs (Table 1). These occurrences were distributed across four regions: Elísio Medrado (n = 1), Camamu (n = 1), Una (n = 1), and within an area of 102 km$^2$ from northern Ilhéus and adjacent municipalities (n = 8; Fig 1).

Among the most recent unpublished records, a specimen was sighted in 2006 by researcher Dr. Priscila Suscke Gouveia (pers. comm., Table 1) in the Una Biological Reserve, Bahia, during her monitoring of capuchin monkeys. This was the first recorded sighting of the painted tree-rat in a strictly protected area. A second specimen was reported in 2011 by an unidentified researcher who observed and captured an individual in a shaded cocoa plantation in the municipality of Ilhéus during the Environmental Impact Study (EIA) for the Porto Sul project (IBAMA 2012) [34]. The specimen has been deposited in the scientific collection of the Universidade Federal da Bahia (Registration number: MZUFBA-649). Finally, a third record corresponds to a female attacked by domestic dogs in 2018 in the municipality of Camamu, Bahia, on the Camamu-Travessão road [35]. This specimen was collected by Dr. Martin R. del Valle Alvarez following information provided by local residents (Alvarez, M.R.V., pers.

**Table 1. Occurrence records of *Callistomys pictus*.**

| Longitude | Latitude | Municipality | Year | Source |
|---|---|---|---|---|
| -39.4815 | -12.872 | Elísio Medrado | 1997 | Encarnação et al. (2000)* [24] |
| -39.2659 | -14.1168 | Camamu | 2018 | This study** |
| -39.1917 | -14.6608 | Ilhéus | ~1990 | Vaz (2002) [25] |
| -39.1671 | -14.6715 | Ilhéus | 2011 | This study*** |
| -39.275 | -14.6822 | Ilhéus | 2003 | Loss et al. (2014) [18] |
| -39.1669 | -14.6838 | Ilhéus | 1944 | Vaz (2002), Ventura et al. (2008) [22] |
| -39.2561 | -14.6961 | Ilhéus | 2002 | Ventura et al. (2008) [22] |
| -39.167 | -14.7005 | Ilhéus | 1986 | Vaz (2002) [25] |
| -39.3 | -14.7333 | Itabuna | 1993 | Vaz (2002) [25] |
| -39.2169 | -14.7671 | Ilhéus | 1997 | Vaz (2002) [25], Encarnação et al. (2000) [24] |
| -39.0528 | -15.1875 | Una | 2006 | This study**** |

Occurrence data for the painted tree-rat (*Callistomys pictus*) compiled from literature reviews, scientific collections, online databases, and consultations with mastozoologists. The locations were sorted in descending order by latitude.

*The originally published coordinates (-39.466667, -12.850000) were inaccurate. The coordinates were set for the shed in Jequitibá Reserve, where the species was sighted according to the published description and personal communication from Marco A. de Freitas.

**Alvarez et al. (2021)

***IBAMA (2012) [34]

****Gouveia P.S., pers. comm.

comm). It has been deposited in the scientific collection "Coleção de Mamíferos Alexandre Rodrigues Ferreira" at the Universidade Estadual de Santa Cruz (Registration number: CMARF—1230). The first and third records confirm the presence of the species in two regions where the painted tree-rat had not been previously documented (Fig 1B).

## Potential occurrence area and climate suitability

The selected ENM (S2 Table) predicted a potential distribution area (climate-suitable area) of 23,151 km$^2$ for the painted tree-rat (Table 2). This area (highlighted in yellow in Fig 2A) is bounded to the east by the ocean and to the north, south, and west by the municipalities of Castro Alves, Belmonte, and Jequié, respectively, in Bahia (38.86–39.97˚W; 12.80–17.03˚S). Additionally, a small area suitable for the climate was identified further south in the municipality of Porto Seguro. The painted tree-rat's potential occurrence area was primarily inserted

**Table 2. Extent of potential occurrence area and core area (km$^2$) for the painted tree rat.**

| | Potential occurrence area | Core area |
|---|---|---|
| Total area (km$^2$) | 23151 | 9225 |
| Total forested area (km$^2$) | 13728 | 6550 |
| Total protected area (km$^2$) | 643 | 243 |
| Total protected forested area (km$^2$) | 602 | 229 |
| Number of municipalities | 76 | 35 |
| Number of protected areas | 9 | 4 |

Extent of potential occurrence area and core area (km$^2$) for the painted tree rat. Within these, the extent of forested area, protected areas, and protected forest area, as well as the number of municipalities, are shown. The potential occurrence area and core area were delineated using binary climate models based on the thresholds of "minimum training presence" and "10th percentile training presence," respectively.

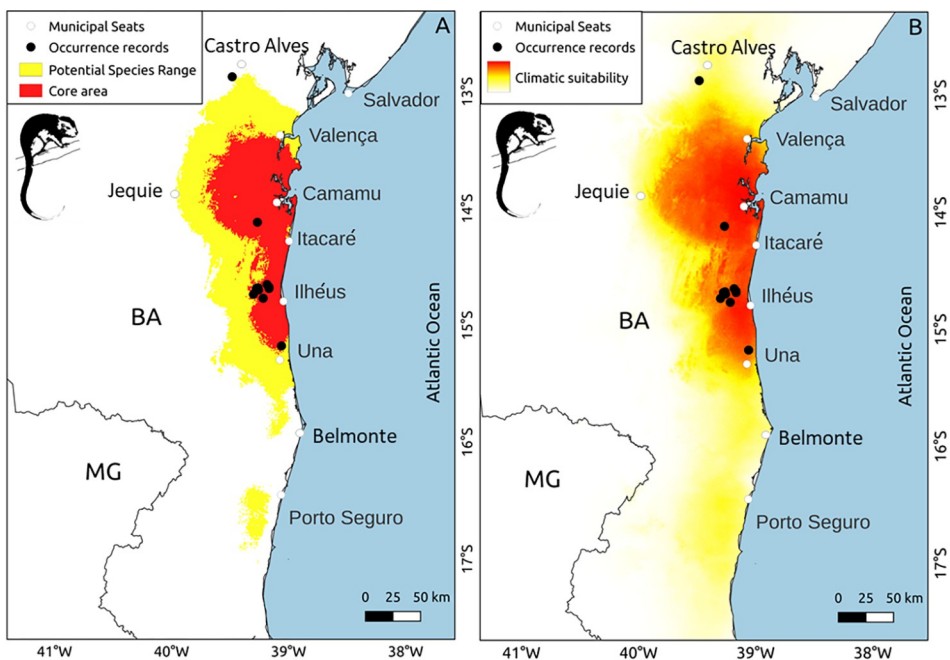

**Fig 2. Potential distribution of *Callistomys pictus*.** (A) Potential distribution area (yellow) and core area (red) of the painted tree-rat (*Callistomys pictus*), as predicted by the Ecological Niche Model. These areas are defined by the "minimum training presence" and "10th percentile training presence" thresholds, respectively. (B) Map showing the gradient of climate suitability for the painted tree-rat (*Callistomys pictus*). The gradient from red to white represents the shift from higher to lower climate suitability for the species. The acronyms represent the names of the Brazilian states: BA—Bahia, MG—Minas Gerais.

(>95%) in the Bahia Coastal Forests ecoregion, known for its evergreen forests. The climate suitability primarily decreases from the coast towards the interior, moving from east to west (see Fig 2B), transitioning from evergreen forests to the drier, seasonal forests.

Approximately 40% (9,225 km², Table 2) of the potential occurrence area was highly suitable (climate suitability ≥ 0.744), encompassing 90% of the known occurrences. This core area (highlighted in red in Fig 2A) is bordered to the east by the ocean, and to the north, south, and west by the municipalities of Valença, Una, and Apuarema (located 38 km east of Jequié), respectively (38.89° to 39.76°W and 13.20° to 15.25°S). This area encompasses parts of the socioeconomic regions in Bahia state known as the Região Cacaueira (Cocoa Region), particularly between Una and Itacaré, and the Costa do Dendê (Palm Oil Coast), stretching from Itacaré to Valença.

The most important bioclimatic variables in predicting range areas for *Callistomys pictus* were Temperature Seasonality (Bio 4), Precipitation Seasonality (Bio 15), Isothermality (Bio 3), and Temperature Annual Range (Bio 7). The contribution ratio (%), permutation importance of each variable can be accessed in S3 Table and in S3 Fig.

## Remaining forested habitat, protected areas, and municipalities

A total of 13,728 km² and 6,550 km² of remaining forested habitat (native forests and shaded cocoa plantations) were predicted within the potential distribution and core area of the painted tree-rat, corresponding to 59% and 71% of these areas, respectively (Table 2). Protected areas encompass 2.8% of the species' potential range and 2.6% of its core area, containing 4.4% and 3.5% of the forested vegetation in these areas, respectively (Table 2). We identified nine strictly

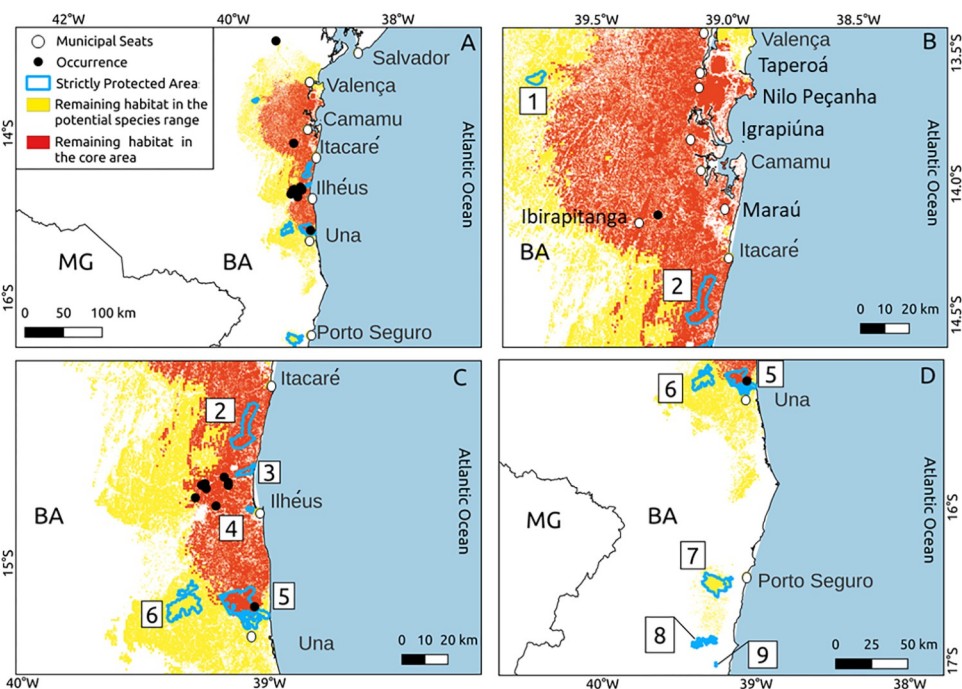

**Fig 3. Forested habitat and protected areas in potential distribution area.** Map of the remaining forested habitat within the potential distribution area (yellow) and core area (red) of the painted tree-rat (*Callistomys pictus*). The map highlights (A) selected municipalities, and (B, C, and D) all strictly protected areas (outlined in blue) within these regions. The identification numbers for the protected areas are: 1 –Wenceslau Guimarães State Ecological Station, 2—Serra do Conduru State Park, 3—Ponta da Tulha State Park, 4—Boa Esperança Municipal Natural Park, 5—Una Biological Reserve, 6—Serra das Lontras National Park, 7—Pau Brasil National Park, 8—Monte Pascoal National Park, and 9—Descobrimento National Park. The acronyms represent the names of the Brazilian states: BA—Bahia, MG—Minas Gerais.

protected areas within the potential range of the painted tree-rat (Fig 3), four of which are located in the core area (Table 3), indicating a higher likelihood of supporting the species. These were situated between Una and Itacaré in the Cocoa region. No strictly protected areas were present between Itacaré and Valença in the Oil Palm Coast region.

We identified 76 municipalities within the potential range of the painted tree-rat, of which 14 had a significant amount of forested area ($> 300$ km$^2$) (Table 4). The core area includes 35 municipalities, nine of which have a large forested area ($> 300$ km$^2$). The municipalities with the largest areas of forested vegetation in the core region, listed in descending order, are: Ilhéus, Camamu, Itacaré, Maraú, Valença, Taperoá, Igrapiúna, Ibirapitanga, and Nilo Peçanha. The species was confirmed to be present only in the first two municipalities.

## Discussion

*Callistomys pictus* has been spotted by researchers on three occasions in the past 20 years, which is both encouraging and concerning. The positive news is that the species still exists in northern Ilhéus and has been discovered in two new locations: the municipality of Camamu and the Una Biological Reserve. Notably, this is the first time the species has been recorded in a strictly protected area. This suggests that the species is not extinct and that new habitats may still be found, including in protected areas. It also confirms that the species is not limited to the area north of Ilhéus and its surroundings. However, the limited number of records and

**Table 3. Protected areas within the potential occurrence area and core area of the painted tree-rat (*Callistomys pictus*).**

| ID[a] | Protected area (Municipality) | Total area (km2) | Climate suitable area (km2) | Forested area (km2) |
|---|---|---|---|---|
| **Potential occurrence area** | | | | |
| 5 | Una Biological Reserve (Una) | 187.25 | 187.25 | 173.16 |
| 7 | Pau Brasil National Park (Porto Seguro) | 189.34 | 176.19 | 161.58 |
| 6 | Serra das Lontras National Park (Arataca) | 113.44 | 113.44 | 106.89 |
| 2 | Serra do Conduru State Park (Itacaré) | 91.51 | 91.51 | 85.89 |
| 8 | Monte Pascoal National Park (Porto Seguro) | 222.39 | 38.11 | 30.19 |
| 1 | W.G. State Ecological Station (W.G.) [a] | 24.19 | 24.19 | 22.2 |
| 3 | Ponta da Tulha State Park (Ilhéus) | 17.05 | 16.94 | 14.65 |
| 4 | Boa Esperança Municipal Park (Ilhéus) | 4.27 | 4.27 | 4.1 |
| 9 | Descobrimento National Park (Prado) | 226.94 | 1.32 | 1.28 |
| **Core area** | | | | |
| 5 | Una Biological Reserve (Una) | 187.25 | 130.29 | 124.88 |
| 2 | Serra do Conduru State Park (Itacaré) | 91.51 | 91.51 | 85.89 |
| 3 | Ponta da Tulha State Park (Ilhéus) | 17.05 | 16.94 | 14.65 |
| 4 | Boa Esperança Municipal N. Park (Ilhéus) | 4.27 | 4.27 | 4.1 |

Protected areas within the potential occurrence area and core area of the painted tree-rat (*Callistomys pictus*). The protected areas were ranked based on the amount of remaining forested area (km$^2$) within their boundaries.

[a]W.G. mean "Wenceslau Guimarães", a municipality from Bahia state.

confirmed locations continues to be a major cause for concern about population declines and extinction risk.

Our predictive ENM indicated that the painted tree-rat has the potential to occur in an area of 23,151 km$^2$ where suitable climatic conditions were predicted for the species. This area is more restricted than the extent of occurrence considered by the IUCN (27,160 km$^2$, [19]). For example, the northern portion of the IUCN's polygon includes areas outside the Atlantic Forest domain (Fig 1), including drier zones (and biomes) and climatic conditions where the species is unlikely to occur. Our model predicted low climatic suitability in drier and seasonal forests, corroborating previous findings that the species' habitat is associated with evergreen forests [23,25]. Therefore, the potential distribution predicted by this study appears to be more realistic than that suggested by the IUCN polygon, and can serve as a better tool to inform conservation policies by refining our understanding of habitat suitability.

We observed low climate suitability in and around Serra da Jibóia (municipality of Elísio Medrado), which is the northernmost limit where the species had previously been reported [24]. In this drier region, the high altitude (reaching up to 820 m) and moist oceanic air masses that reach the Serra da Jibóia allow the existence of ombrophilous dense forest in its eastern portion [52], likely providing some climatic suitability for the species. However, if our predictions are accurate, this area represents a climatic anomaly within the regional context, likely harboring an isolated population, as the surrounding lowland areas appear to have unsuitable climate conditions and are heavily deforested (Fig 3B). Future field surveys and genetic studies may be conducted in this region to assess the status of this population.

We identified a core area of 9,225 km$^2$ along the coast between Una and Valença, which is highly likely to be inhabited by the species. In this section, we emphasize that certain municipalities and protected regions require increased focus from researchers and conservationists. Firstly, given the limited number of confirmed habitats, actively protecting all known populations is crucial for the conservation of the species. We focus on the unprotected and four protected areas of Ilhéus and neighboring municipalities such as Una and Itacaré. This is because

**Table 4. Municipalities with forested areas.**

| Municipality | Climate suitable area (km$^2$) | Forested area (km$^2$) |
|---|---|---|
| **Potential occurrence area** | | |
| Ilhéus | 1830.01 | 1421.08 |
| Una | 1151.33 | 901.18 |
| Valença | 1156.16 | 711.41 |
| Itacaré | 730.86 | 604.22 |
| Camamu | 817.18 | 577.25 |
| Maraú | 739.05 | 497.74 |
| W.G.[a] | 662.34 | 409.57 |
| Santa Luzia | 586 | 391.85 |
| Porto Seguro | 846.21 | 371.46 |
| Arataca | 393.81 | 347.15 |
| Ibirapitanga | 469.85 | 329.54 |
| Taperoá | 405.29 | 309.35 |
| Igrapiúna | 443.66 | 308.02 |
| Nilo Peçanha | 385.4 | 303.88 |
| **Core area** | | |
| Ilhéus | 1419.97 | 1125.48 |
| Camamu | 817.18 | 577.25 |
| Itacaré | 589.25 | 488.25 |
| Maraú | 715.16 | 483.75 |
| Valença | 433.37 | 310.04 |
| Taperoá | 405.29 | 309.35 |
| Igrapiúna | 443.66 | 308.02 |
| Ibirapitanga | 438.64 | 307.83 |
| Nilo Peçanha | 385.4 | 303.88 |

Municipalities with the largest forested areas within the potential and core habitats of the painted tree rat (*Callistomys pictus*). The municipalities were ranked according to the amount of remaining forested area (km$^2$) within their limits.

[a]W.G. mean "Wenceslau Guimarães", a municipality from Bahia state.

most of the known specimens (nine out of 11) originate from this region, and these municipalities, along with the protected areas, exhibit the highest climate suitability and the greatest extent of forested vegetation within the potential range of the species.

The confirmed presence of the species in the Una Biological Reserve is indeed promising news, as this reserve spans approximately 173.16 km$^2$ of continuous forested environment, with 72.1% falling within the predicted core area of the species. Furthermore, there are at least two other significant protected areas within this region, Serra do Conduru State Park and Serra das Lontras National Park, which could potentially serve as habitats for this species. However, confirmation is required as this species has not been previously reported in these two protected areas [53–55].

Our model predicted a large area of suitable climate between Itacaré and Valença (in the Oil Palm Coast socioeconomic region), which corresponds to the location of the most recent record of the species. Some studies have identified that this region is important for the conservation of several threatened arboreal mammals, such as the maned sloths, thin-spined porcupines, and primates [56–58]. However, there are no strictly protected areas in this region. Given its potential importance and extensive forest cover, we emphasize the need for increased

survey efforts to confirm the presence of the painted tree-rat in this region. If new populations are confirmed, conservation measures may be tailored to the local context. We identify Camamu, Itacaré, Maraú, Valença, Taperoá, Igrapiúna, Ibirapitanga, and Nilo Peçanha as the municipalities most likely to harbor the species in this region, where future surveys should be prioritized.

Detecting this species in the field is apparently a great challenge. All the specimens were recently detected through opportunistic sightings. The painted tree-rat has never been detected by traditional mammal survey techniques (e.g., live trapping, camera trapping, active search, and line transect surveys) during sampling carried out in the Cocoa and Oil Palm Coast regions over the past three decades [53,54,59–71]. The species was not captured even during the surveys conducted in forest fragments, such as the Una Biological Reserve, where their presence is now confirmed [54,60–62,68,69]. It is likely that these animals, being arboreal and nocturnal, have elusive habits and are not attracted to bait, which makes detecting them difficult. On the other hand, although recognition of this species by local residents has generally been rare during interviews conducted in these regions ([55,67]; Flesher K., personal communication), 56% of local residents recognized the species in the Una Biological Reserve [72], confirming its presence there. In this context, gathering information from residents through interviews and promoting citizen science (e.g., using social networks) may be a cost-effective alternative for discovering new populations, or even learning from local residents how to more efficiently locate the species in the field.

Considering that species not observed tend to receive less conservation attention and study, we intend to shed some light on these "ghost species" to support advances in our understanding of them. This is the first study to assess the potential distribution of the painted tree-rat (*Callistomys pictus*) using Ecological Niche Modeling. Given the small size of the dataset (n = 11), we caution that our model should not be viewed as defining the definitive limits of the species' distribution, but rather as an initial approximation that aids in making informed decisions based on the best available knowledge about the distribution of this endangered species. We have determined that the coastal forest habitat is more suitable for the species. Currently, the representation of this suitable habitat within protected areas is very low. Additionally, there is a high potential for the species to occur in 9 municipalities and four protected areas. Future research in these areas is recommended to confirm its presence and inform conservation strategies.

While new data are not available, the existing records confirm that Ilhéus and surrounding municipalities, including Una, play a significant role in the conservation of the painted tree-rat. In the Ilhéus region, local residents suggest that the species' populations were drastically reduced by the implementation and expansion of cocoa plantations [24]. Given that the species is strictly arboreal [20], it is likely that *C. pictus* continues to be impacted by the loss of native forests and the intensification of management in shade cacao plantations in this region. This intensification typically involves reducing the density of shade trees, canopy cover, and connectivity [73]. Such intensification has impacted the biodiversity and habitats of various forest species groups in the Cocoa region [61,69,74–77]. Intuitively, enhancing or preserving the extent, connectivity, and complexity of these forested environments (forests and agroforests) would be a logical strategy to promote the conservation of this arboreal species. Thus, actions that promote the conservation and restoration of these habitats are recommended, such as expanding the network of protected areas (including private ones), restoring forests, controlling deforestation, providing environmental education, and promoting alternative practices (e.g., ecotourism, payment for environmental services) that add value for cocoa farmers who maintain native forests and the structural complexity of their plantations (see [54,61,73]). Promoting high density of shade trees, abundance of vines, and biodiversity-friendly management

practices in cacao-forest systems is likely to benefit *C. pictus* populations. Clearly, conservation of this species will also depend on our ability to detect and monitor its populations, understand its habitat requirements, and preserve the integrity of its habitat through coordinated efforts.

## Supporting information

**S1 Table. Occurrence data of the painted tree-rat (*Callistomys pictus*) recorded to date.** These were documented in the literature, deposited in scientific collections, and/or observed by mastozoologists. Data entries with geographical coordinates were sorted by latitude, while those without coordinates were sorted by year. [a] asterisk indicate the holotype; abbreviations are: RM233- field code reported in Loss et al. (2014); w/n -"without number"; CMARF– Coleção de Mamíferos Alexandre Rodrigues Ferreira, Universidade Estadual de Santa Cruz, Ilhéus, Brazil; MZUFBA—Museu de Zoologia, Universidade Federal da Bahia, Salvador, Brazil; MN—Museu Nacional do Rio de Janeiro, Universidade Federal do Rio de Janeiro, Rio de Janeiro, Brazil; MZUSP—Museu de Zoologia, Universidade de São Paulo, São Paulo, Brazil; NHM: Natural History Museum—London, England; MHNN—Museum d'Histoire Naturelle du Neuchatel, Neuchatel, Switzerland; MNK—Museum für Naturkunde, Berlin, Germany; CEPEC: Centro de Pesquisas do Cacau, Comissão Executiva do Plano da Lavoura Cacaueira, Ilheus, Brazil. [b] abreviations: ske—skeleton, ski–skin; sku–skull; mski—mounted skin; alc– specimen preserved in alcohol. [c] references: 1—Encarnação et al. (2000) [24]; 2—Alvarez et al. (2021); 3—Vaz (2002) [25]; 4—IBAMA (2012) [34]; 5—Loss et al. (2014) [18]; 6—Ventura et al. (2008) [22]; 7—Gouveia P.S., pers. comm.; 8 –GBIF database. (XLSX)

**S2 Table. Parameters and metrics of the top 10 distribution models for the painted tree-rat (*Callistomys pictus*).** Only models with Δ.AICc < 2 are shown. The models were constructed using all 15 possible combinations of four feature classes (Linear = L, Quadratic = Q, Product = P, and Hinge = H) along with various regularization multipliers (ranging from 0.5 to 4.5 in increments of 0.5). The significance of the model (partial ROC, p-value), omission rates for two climate suitability thresholds (ORmtp and OR10), and Akaike's Information Criterion (AICc) are presented. (XLSX)

**S3 Table. Contribution ratio (%) and permutation importance of each bioclimatic variable used to model the potential distribution of *Callistomys pictus*.** The values are based on the single "best" model. (XLSX)

**S1 Fig. Correlation between pairs of bioclimatic variables calculated using the Pearson Coefficient.** Variables with Pearson's correlation coefficients below 0.70 were selected: Annual mean temperature (Bio1), isothermality (Bio3), temperature seasonality (Bio4), temperature annual range (Bio7), precipitation of the wettest month (Bio13), and precipitation seasonality (Bio15). (TIF)

**S2 Fig.** Comparison of the results from the single "best" model (A and B) and the ensemble model (C and D). The single "best" model is selected based on the lowest AIC value, while the ensemble model is based on the average values from the 10% of models with the lowest AIC values. A and C show the climate suitability gradient for the painted tree-rat (*Callistomys pictus*) as predicted by the best continuous model and the ensemble continuous model, respectively. B and D depict the core areas (highlighted in green) defined by the "10th percentile

training presence" threshold. In the continuous models, the gradient from red to blue represents the shift from higher to lower climate suitability for the species.
(TIF)

**S3 Fig. Response curves of the variables with the greatest effect on the output of the model of current conditions.**
(TIF)

## Acknowledgments

We are grateful to the Applied Ecology & Conservation Lab and Santa Cruz State University for providing infrastructure, logistical, and academic support. We are grateful to Priscila S. Gouveia, Antônio J. S. Argôlo, Marco A. de Freitas, Kevin Flesher, and Martin R. V. Alvarez for providing information and helping to validate the geographic coordinates of recently observed specimens.

## Author Contributions

**Conceptualization:** Andrés David Sarmiento Sanchez, Gastón Andrés Fernandez Giné.

**Data curation:** Andrés David Sarmiento Sanchez, Gastón Andrés Fernandez Giné.

**Formal analysis:** Andrés David Sarmiento Sanchez, Gabriela Alves-Ferreira, Gastón Andrés Fernandez Giné.

**Funding acquisition:** Gastón Andrés Fernandez Giné.

**Investigation:** Andrés David Sarmiento Sanchez, Neander Marcel Heming, Gastón Andrés Fernandez Giné.

**Methodology:** Andrés David Sarmiento Sanchez, Gabriela Alves-Ferreira, Neander Marcel Heming, Gastón Andrés Fernandez Giné.

**Project administration:** Gastón Andrés Fernandez Giné.

**Resources:** Gastón Andrés Fernandez Giné.

**Software:** Gastón Andrés Fernandez Giné.

**Supervision:** Gastón Andrés Fernandez Giné.

**Validation:** Gabriela Alves-Ferreira, Neander Marcel Heming, Gastón Andrés Fernandez Giné.

**Visualization:** Andrés David Sarmiento Sanchez, Gabriela Alves-Ferreira, Neander Marcel Heming, Gastón Andrés Fernandez Giné.

**Writing – original draft:** Andrés David Sarmiento Sanchez, Gastón Andrés Fernandez Giné.

**Writing – review & editing:** Andrés David Sarmiento Sanchez, Gabriela Alves-Ferreira, Neander Marcel Heming, Gastón Andrés Fernandez Giné.

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
