## [Decision Letter · Decision Letter 0]

29 Oct 2024

PONE-D-24-33359Distribution and Habitat of the Painted Tree Rat (Callistomys pictus): Evaluating Areas for Future Surveys and Conservation EffortsPLOS ONE

Dear Dr. Giné,

Thank you for submitting your manuscript to PLOS ONE. After careful consideration, we feel that it has merit but does not fully meet PLOS ONE’s publication criteria as it currently stands. Therefore, we invite you to submit a revised version of the manuscript that addresses the points raised during the review process.

We look forward to receiving your revised manuscript.

Kind regards,

Daniel de Paiva Silva, Ph.D.

Academic Editor

PLOS ONE

**Journal Requirements:**

ADSS received a fellowship from Fundação de Amparo à Pesquisa do Estado da Bahia (FAPESB) during the preparation of this study.

https://www.fapesb.ba.gov.br/

GAF received a doctoral scholarship from Coordenação de Aperfeiçoamento de Pessoal de Nível Superior during the preparation of this study (code 001).

https://www.gov.br/capes/pt-br

3. We note that Figures 1, 2 and 3 in your submission contain map images which may be copyrighted. All PLOS content is published under the Creative Commons Attribution License (CC BY 4.0), which means that the manuscript, images, and Supporting Information files will be freely available online, and any third party is permitted to access, download, copy, distribute, and use these materials in any way, even commercially, with proper attribution. For these reasons, we cannot publish previously copyrighted maps or satellite images created using proprietary data, such as Google software (Google Maps, Street View, and Earth). For more information, see our copyright guidelines: http://journals.plos.org/plosone/s/licenses-and-copyright.

We require you to either present written permission from the copyright holder to publish these figures specifically under the CC BY 4.0 license, or remove the figures from your submission:

a. You may seek permission from the original copyright holder of Figures 1, 2 and 3 to publish the content specifically under the CC BY 4.0 license.  

**Additional Editor Comments:**

Dear Dr. Giné,

After this first round of reviews, both reviewers believe our manuscript may be accepted for publication after major reviews are applied. Please consider all suggestions made by the reviewers and provide a rebuttal letter inform whether the change was made or justifying why the change was not performed.

Sincerely,

Daniel Silva

Reviewers' comments:

Reviewer's Responses to Questions

**Comments to the Author**

1. Is the manuscript technically sound, and do the data support the conclusions?

Reviewer #1: Yes

Reviewer #2: Yes

2. Has the statistical analysis been performed appropriately and rigorously? 

Reviewer #1: Yes

Reviewer #2: Yes

3. Have the authors made all data underlying the findings in their manuscript fully available?

Reviewer #1: Yes

Reviewer #2: Yes

4. Is the manuscript presented in an intelligible fashion and written in standard English?

Reviewer #1: Yes

Reviewer #2: Yes

5. Review Comments to the Author

**Reviewer #1:** Manuscript Number: PONE-D-24-33359

"Distribution and Habitat of the Painted Tree Rat (Callistomys pictus): Evaluating Areas

for Future Surveys and Conservation Efforts"

Comments to Author (also see the attached pdf file):

The manuscript deals with ecological niche modeling of current distribution of painted tree-rat (Callistomys pictus). Standard bioclimatic predictors from the WorldClim database and the species occurrence data according to the global biodiversity information and other published sources have been used to assess the potential distribution of the species using the maximum entropy modeling algorithm for the first time. Hence, the aim of the paper is rather ambitious.

Overall, the paper is concise and quiet well written, reasonably clear and the range of analytical methods are appropriate and well used. Statistical analysis is well described and conducted rigorously. However, I honestly think that basing such a study on a very small dataset with few point localities for the species can be dangerous. I suppose that less than at least 30 points cannot be representative of the distribution of a species at a national scale, nor trustable. Anyway, with regards to the author explanations throughout the text and the nature of the species, one can accept such an analysis for future surveys and conservation efforts of a rare unknown animal.

It should be considered that the presence-only data sets often suffer from spatiotemporal autocorrelation and therefore, potentially introduce environmental bias into modeling.

To address the issues related to the use of occurrence-only data in ecological modeling, I have the following suggestion which need authors’ attention: to correct sampling bias, it is important to spatially rarefy data depending on habitat and climatic heterogeneity to minimize environmental bias and to make it spatially independent. This is done with the rarefying tool available in the ArcGIS toolbox implemented in ArcMap and accounting sampling bias with a bias grid (I can refer authors to e.g., Brown, 2014. SDM toolbox: a python-based GIS toolkit for landscape genetic, biogeographic and species distribution model analyses. Methods in Ecology and Evolution, 5, 694–700. https://doi.org/10.1111/2041-210X.12200).

I would also suggest showing the bioclimatic variables used to develop the distribution model in Maxent as a new table, along with the contribution ratio (%) and permutation importance value for each layer. Moreover, the response curves of the variables with the greatest effect on the output of the model of current conditions can be added as the supplementary material.

I have also minor comments to do (also see the attached pdf file):

1. Please ensure that your manuscript meets PLOS ONE's style requirements.

2. I would like to recommend an English review since there are some small punctuation mistakes and not clear/long sentences, which sometimes are difficult to understand.

3. References seems to be a mixture of different formats and in somehow, not well addressed.

Finally, my recommendation for this manuscript, is “Accept after major revision”.

**Reviewer #2: **The study offers valuable insights into the potential range of this endangered species, providing a foundation for future fieldwork and conservation strategies in the Atlantic Forest of Brazil. The manuscript is well-structured and addresses a significant knowledge gap regarding the painted tree rat, a species with very limited recent sightings.

My general suggestions:

The title of the article mention both recommendations for future surveys and conservation efforts. The authors did a great job discussing priorities for future surveys, but the recommendations for conservation efforts could be further developed. The authors described the strictly-protected areas within the suitable areas for the species, but where specifically should conservation efforts be focused? For example, where new conservation areas would be most benefitting for the species based on your results?

If there is still time for that, I think it would be beneficial to include non-strictly protected areas in the analysis. They also can play an important role in the conservation of small mammal species. I don't think they should be clumped with the strictly protected areas, but represent another level of protection in Table 2. If you prefer to not include non-strictly protected areas, I suggest you to add your reason to the methods section.

Specific comments:

Line 95: I think the authors need to clarify what is an unpublished location and where grey literature publications stand in their definition of unpublished location.

Line 97: I think more context is needed for question 3 as well. New surveys and conservation efforts are two very different actions with different purposes. It is confusing to have them together without context. Reading the rest of your paragraph, I noticed that you suggested areas with high forest cover and climate suitability as areas of high priority for new surveys. This is good, but what about the conservation efforts that you mentioned in question 3? I suggest you to: (1) make question 3 about priority areas for new surveys; (2) complement the text with the benefit of targeted new surveys [would them increase the likelihood that the species would be successfully surveyed?]; [3] complement the text with the benefit of improving the knowledge about the species with new surveys; [4] if you want to keep the part about "where conservation efforts should be focused", I recommend adding morre context about it. How you are defining conservation priority? What is the benefit of protecting those areas for the species. Let me give you an example: we could suggest that conservation efforts are focused in areas of high climate suitability, but low forest cover, because there the populations could be under higher risk. This would be a reactive conservation action. Or, conservation efforts could be focused in areas of high climate suitability but high forest cover, to protect the "healthier" populations of the species. There is no need to write about those differences in your text, but it is important to make it explicit about how you are informing conservation efforts with your research.

Line 100: what was included in literature? what type of reports? Provide a simplified list, for example, peer-reviewed articles, public dissertations, xxxx...

Line 101: increment the text with: "consultation with mastozoologists working in the region"

Introduction: You have a co-dependency between your objectives/questions. Objective/question one needs to answered to provide data to objective/question 2. Objective/question 2 needs to answered to provide necessary information to answer question/objective 3. And objective 3 is important to provide information that can be used to guide policy that can support the survival of this species. I suggest the authors to take advantage of this co-dependent research structure when writing the last paragraph of the Introduction.

Methods: I suggest that the methods start with question 1 of your Introduction. How did you answer question 1? That should be the first method appearing. By starting the methods section saying that you assessed the potential distribution, you confuse the reader about the objectives of the study. I suggest to start with "Study area description", followed by how you answered question 1: "Through data compilation", then going to question 2: "environmental variables" and "niche modelling".

Line 131: this section gained importance, and needs to be detailed, because finding new records is your objective 1. What was considered "literature" and "grey-literature"? What types of report were considered? EIA reports? were thesis and dissertation considered?

Line 143: when you mentioned that you did not consider data from interviews, what does that mean? Newspaper interview? Wouldn't be worth to consult with the interviewed author?

Line 149: Delete the phrase: "We used this compiled dataset to construct the models.". This can be added later when you are talking about models

Line 151: "We downloaded data for 19 bioclimatic variables". For which year? I see that you have occurrences that go back to 1944, so I am curious to know what is the time extent of your bioclimatic variables. No need to change the analysis, but make it explicit.

Line 155: The Pearson correlation coefficients were not made available in the supplementary material. I suggest the authors to add it there for transparency.

Line 157: Substitute "used" by "selected"

Method: I wonder the impact of the calibration extent in your results. It can be worth to describe why to use the 155 km threshold from the MCP. The concern is that detectability for this species might be so low, that the distribution could go much further than the calibration extent. This is not a suggestion, but just something worth a small explanation about the choice of calibration area.

Line 227: Describe what more precise provenance means in the text.

Results: Your first question is ": Have there been any new, unpublished locations where the species has been found? ". Line 242 you are describing what it seems to be these new occurrences, make it more explicit in the beginning of the paragraph that you are answering question 1. Were these records obtained through reports or consultation with specialist. This is not clear for the occurrence taken by Dr. Gouveia, but it is clear in the second EIA report example. Make it clear for all three records.

Line 233: Can you mention in table 1 the type of source, which ones were compiled by consultation, etc, etc?

Line 288: Why non-strictly protected areas were not considered? They can also be important for the conservation of small mammals, and the information about their cover within the suitable areas for the cocoa tree-rat could be useful for conservation policies for the species

Table 4. The legend of table 4 is very superficial in its current form. I suggest the authors to do something similar to Table 3, and describe what the reader is seeing in the table. It is not possible to know what table 4 is about just by reading : Municipalities with forested area.

Line 345: Complete the phrase: "However, the limited number of records and confirmed locations continues to be a major cause for concern" with what is the concern about, extinction risk, lack of knowledge, population decline, etc.

Line 346: Here it is worth to mention how the IUCN calculates the extent of occurrence. You could briefly mention how Ecological niche modelling, by being more realistic, can offer an useful tool to inform conservation policies. I imagine that you are going to talk more about it later in the Discussion, but it could be helpful to start the conversation here.

Line 440: This paragraph is great. I would reinforce the benefit of non-managed cacao-forest systems for this species (this is briefly mentioned in line 429, but it could make to the list of actions to support C. pictus populations).

6. PLOS authors have the option to publish the peer review history of their article (what does this mean?). If published, this will include your full peer review and any attached files.

Reviewer #1: No

Reviewer #2: No

---

## [Author Response · Author response to Decision Letter 0]

17 Dec 2024

Dear Phd Daniel de Paiva Silva,

Academic Editor

PLOS ONE

Ilhéus -Brazil, December 5, 2024

We sincerely thank you for the opportunity to resubmit our manuscript, “Distribution and Habitat of the Painted Tree Rat (Callistomys pictus): Evaluating Areas for Future Surveys and Conservation Efforts” to Plos One. The feedback of the reviewers and editor was highly constructive and has greatly enhanced the quality of the manuscript. We have addressed each comment below and provided a point-by-point response to each comment. The main changes in this version of the manuscript include: (1) We added a supplementary table with the contribution ratio (%) and permutation importance for each bioclimatic variable used in the models; (2) we ensured the manuscript meets PLOS ONE style requirements, (3) clarified what is an unpublished location and where grey literature publications stand in their definition of unpublished location; (4) we modified the objectives as suggested by the reviewer 2; (5) we tested the method suggested by the reviewer to filtering the occurrence records. The minor changes can be seen in detail in the manuscript and in the response letter. If you have any questions, please contact me.

Kind regards,

Gastón A. F. Giné,

Universidade Estadual de Santa Cruz, Brazil

Response to Editor: 

Reply: We have ensured that the manuscript complies with the PLOS ONE style requirements.

ADSS received a fellowship from Fundação de Amparo à Pesquisa do Estado da Bahia (FAPESB) during the preparation of this study.

https://www.fapesb.ba.gov.br/

GAF received a doctoral scholarship from Coordenação de Aperfeiçoamento de Pessoal de Nível Superior during the preparation of this study (code 001).

https://www.gov.br/capes/pt-br

Reply: Thank you, we added this statement. We fixed for “ADSS received a fellowship from Conselho Nacional de Desenvolvimento Científico e Tecnológico (CNPq) during the preparation of this study.”

3. We note that Figures 1, 2 and 3 in your submission contain map images which may be copyrighted. All PLOS content is published under the Creative Commons Attribution License (CC BY 4.0), which means that the manuscript, images, and Supporting Information files will be freely available online, and any third party is permitted to access, download, copy, distribute, and use these materials in any way, even commercially, with proper attribution. For these reasons, we cannot publish previously copyrighted maps or satellite images created using proprietary data, such as Google software (Google Maps, Street View, and Earth). For more information, see our copyright guidelines: http://journals.plos.org/plosone/s/licenses-and-copyright.

We require you to either present written permission from the copyright holder to publish these figures specifically under the CC BY 4.0 license, or remove the figures from your submission: a. You may seek permission from the original copyright holder of Figures 1, 2 and 3 to publish the content specifically under the CC BY 4.0 license. 

Reply: Figures 1, 2, and 3 were produced by us, the authors, using the open-source software QGIS, with shapefiles obtained from databases licensed under the CC BY 4.0 or similarly permissive licenses. The data sources include:

Brazilian Government Open Data Portal: Conservation Units Dataset. Available at: https://dados.gov.br/dados/conjuntos-dados/unidadesdeconservacao

Natural Earth: Free Vector and Raster Map Data. Available at: https://www.naturalearthdata.com

Terra brasilis. https://terrabrasilis.dpi.inpe.br/downloads/

These data sources allow free use, distribution, and modification.

Reviewer #1: Manuscript Number: PONE-D-24-33359

"Distribution and Habitat of the Painted Tree Rat (Callistomys pictus): Evaluating Areas

for Future Surveys and Conservation Efforts"

Comments to Author (also see the attached pdf file):

The manuscript deals with ecological niche modeling of current distribution of painted tree-rat (Callistomys pictus). Standard bioclimatic predictors from the WorldClim database and the species occurrence data according to the global biodiversity information and other published sources have been used to assess the potential distribution of the species using the maximum entropy modeling algorithm for the first time. Hence, the aim of the paper is rather ambitious. Overall, the paper is concise and quiet well written, reasonably clear and the range of analytical methods are appropriate and well used. Statistical analysis is well described and conducted rigorously. However, I honestly think that basing such a study on a very small dataset with few point localities for the species can be dangerous. I suppose that less than at least 30 points cannot be representative of the distribution of a species at a national scale, nor trustable. Anyway, with regards to the author explanations throughout the text and the nature of the species, one can accept such an analysis for future surveys and conservation efforts of a rare unknown animal. 

It should be considered that the presence-only data sets often suffer from spatiotemporal autocorrelation and therefore, potentially introduce environmental bias into modeling. To address the issues related to the use of occurrence-only data in ecological modeling, I have the following suggestion which need authors’ attention: to correct sampling bias, it is important to spatially rarefy data depending on habitat and climatic heterogeneity to minimize environmental bias and to make it spatially independent. This is done with the rarefying tool available in the ArcGIS toolbox implemented in ArcMap and accounting sampling bias with a bias grid (I can refer authors to e.g., Brown, 2014. SDM toolbox: a python-based GIS toolkit for landscape genetic, biogeographic and species distribution model analyses. Methods in Ecology and Evolution, 5, 694–700. https://doi.org/10.1111/2041-210X.12200).

Reply: Thank you for the suggestion. We applied the method of spatially rarefying data based on habitat and climatic heterogeneity using the SDMtoolbox tool in ArcGIS. We used the variables bio4, bio15, bio3, and bio7 and distances of 1, 10, and 20 km. When filtering with a distance of 1 km and environmental variables, we obtained the same 11 records that had already been previously modeled. When filtering with environmental variables and 10 km, we obtained 6 occurrence records, and when filtering with environmental variables and 20 km, all occurrence records were eliminated. We redid the niche model using the 6 records obtained through the filtering, as suggested. However, these models showed worse omission rate than those reported with the models we are already using (Table 1). Furthermore, the spatial patterns were very similar to those obtained in our models with 11 occurrence points (Figure 1), but the climatically suitable area does not include and is further away from the northernmost record. Therefore, we decided to keep the models with 11 points in our study. We added “ To reduce the sampling bias, we removed occurrences that were at least 1 km apart.”

Figure 1. Models with 11 occurrence points (a) and models with six occurrence points filtered with environmental variables and distance of 10 km using SDMtoolbox (b).

Table 1. Parameters and metrics of the best model for the painted tree-rat (Callistomys pictus) with 11 and six occurrence records. We selected the "best" model based on its AUC value, partial ROC, lower omission rate (OR), and lower Akaike Information Criterion (AICc) values, following the methodology described in Cobos et al. (2019). The significance of the model (partial ROC, p-value), omission rates for two climate suitability thresholds (ORmtp and OR10), and Akaike's Information Criterion (AICc) are presented. 

Figure 1 and Table 1 can be viewed in the file "Response to Reviewers" that we submitted along with the revised manuscript.

I would also suggest showing the bioclimatic variables used to develop the distribution model in Maxent as a new table, along with the contribution ratio (%) and permutation importance value for each layer. Moreover, the response curves of the variables with the greatest effect on the output of the model of current conditions can be added as the supplementary material.

Reply: Thank you for the excellent suggestion! We added a supplementary table with the contribution ratio (%) and permutation importance for each bioclimatic variable used in the models. We also added the response curves in the supplementary material.

I have also minor comments to do (also see the attached pdf file):

1. Please ensure that your manuscript meets PLOS ONE's style requirements.

2. I would like to recommend an English review since there are some small punctuation mistakes and not clear/long sentences, which sometimes are difficult to understand.

3. References seems to be a mixture of different formats and in somehow, not well addressed.

Reply: Sorry for that, we have formatted the manuscript according to the Plos One style requirements, including the references. We also submitted the manuscript to an english review.

Reviewer #2: The study offers valuable insights into the potential range of this endangered species, providing a foundation for future fieldwork and conservation strategies in the Atlantic Forest of Brazil. The manuscript is well-structured and addresses a significant knowledge gap regarding the painted tree rat, a species with very limited recent sightings. My general suggestions:

The title of the article mentions both recommendations for future surveys and conservation efforts. The authors did a great job discussing priorities for future surveys, but the recommendations for conservation efforts could be further developed. The authors described the strictly-protected areas within the suitable areas for the species, but where specifically should conservation efforts be focused? For example, where new conservation areas would be most benefitting for the species based on your results?

Reply: We chose to be cautious regarding the recommendation of new areas for conservation because, even though the model indicates new climatically suitable areas for the species, this does not mean that the species is actually present in those areas. This precaution is reflected in the sixth paragraph of the discussion, where we stated the following: “If new populations are confirmed, conservation measures may be tailored to the local context.” In the same paragraph, we indicated that the Oil Palm Coast socioeconomic region is a potential geographic zone where conservation efforts should be focused if and where the species is confirmed, based on the low representativity of protected areas in the region. Conservation efforts can vary depending on the local context. Additionally, in the final paragraph of the discussion, we cautiously recommended conservation actions for the region where the presence of the species has been repeatedly confirmed, which is predominantly unprotected and occupied by cocoa cultivation. We concluded the article with various recommendations that could be important for the species in this region.

If there is still time for that, I think it would be beneficial to include non-strictly protected areas in the analysis. They also can play an important role in the conservation of small mammal species. I don't think they should be clumped with the strictly protected areas, but represent another level of protection in Table 2. If you prefer to not include non-strictly protected areas, I suggest you add your reason to the methods section.

Reply: Given the low efficiency of management and conservation of non-strictly protected areas, we decided to be more conservative by highlighting only those that have strict protection. We have added this sentence in the methods.

Specific comments:

Line 95: I think the authors need to clarify what is an unpublished location and where grey literature publications stand in their definition of unpublished location. 

Reply: Unpublished locations are those that have not yet been published in peer-reviewed journals. We obtained these recent locations in gray literature (including Environmental Impact Study (EIA) reports, Brazilian Red Lists, and management plans of protected areas), through personal communication with mastozoologists working in the region, and scientific collections. This is now better detailed in the methodology.

Line 97: I think more context is needed for question 3 as well. New surveys and conservation efforts are two very different actions with different purposes. It is confusing to have them together without context. Reading the rest of your paragraph, I noticed that you suggested areas with high forest cover and climate suitability as areas of high priority for new surveys. This is good, but what about the conservation efforts that you mentioned in question 3? I suggest you to: (1) make question 3 about priority areas for new surveys; (2) complement the text with the benefit of targeted new surveys [would them increase the likelihood that the species would be successfully surveyed?]; [3] complement the text with the benefit of improving the knowledge about the species with new surveys; [4] if you want to keep the part about "where conservation efforts should be focused", I recommend adding morre context about it. How you are defining conservation priority? What is the benefit of protecting those areas for the species. Let me give you an example: we could suggest that conservation efforts are focused in areas of high climate suitability, but low forest cover, because there the populations could be under higher risk. This would be a reactive conservation action. Or, conservation efforts could be focused in areas of high climate suitability but high forest cover, to protect the "healthier" populations of the species. There is no need to write about those differences in your text, but it is important to make it explicit about how you are informing conservation efforts with your research.

Reply: Thank you for the suggestion. 

We modified question 3 as suggested, adding the following sentences: “(3) identify regions, municipalities, and protected areas with the highest potential to harbor the species (based on climate suitability and forest cover) to delineate areas where the species is most likely to occur, guiding future survey efforts”. 

We could make a series of recommendations, such as restoration in climatically suitable areas with low forest cover and habitat conservation in regions with higher forest cover. However, we prefer to be cautious and not proceed with such specific recommendations, as there is no evidence of the species' occurrence in most of the predicted potential distribution, as previously mentioned. Considering the limitations of proposing priority areas for conservation based solely on ENMs before confirming the species' presence, we recommended conservation actions only in areas with repeated confirmations of species occurrence, as well as we highlighted regions with low representativeness of protected areas, which, once the species' presence is confirmed, may receive conservation actions. Therefore, we added the following sentences at the end of the introduction: “Finally, aiming to provide data to future support conservation policies for the target species, we recomm

---

## [Decision Letter · Decision Letter 1]

27 Dec 2024

Distribution and Habitat of the Painted Tree Rat (Callistomys pictus): Evaluating Areas for Future Surveys and Conservation Efforts

PONE-D-24-33359R1

Dear Dr. Giné

We’re pleased to inform you that your manuscript has been judged scientifically suitable for publication and will be formally accepted for publication once it meets all outstanding technical requirements.

Kind regards,

Daniel de Paiva Silva, Ph.D.

Academic Editor

PLOS ONE

Additional Editor Comments (optional):

Dear Dr. Giné,

I am pleased to accept your manuscript for publication in PLoS One! Congratulations!

Sincerely,

Daniel Silva

Reviewers' comments:

Reviewer's Responses to Questions

**Comments to the Author**

1. If the authors have adequately addressed your comments raised in a previous round of review and you feel that this manuscript is now acceptable for publication, you may indicate that here to bypass the “Comments to the Author” section, enter your conflict of interest statement in the “Confidential to Editor” section, and submit your "Accept" recommendation.

Reviewer #1: All comments have been addressed

2. Is the manuscript technically sound, and do the data support the conclusions?

Reviewer #1: Yes

3. Has the statistical analysis been performed appropriately and rigorously? 

Reviewer #1: Yes

4. Have the authors made all data underlying the findings in their manuscript fully available?

Reviewer #1: Yes

5. Is the manuscript presented in an intelligible fashion and written in standard English?

Reviewer #1: Yes

6. Review Comments to the Author

Reviewer #1: After the revision made by the authors, manuscript has improved and some parts which already need explanations, are more clear now, and easy to understand. There are still two more points I would like to mention again:

1- Some of the comments which were shown as sticky notes in the review file (pdf) are not applied (e.g., comments No. 1, 2, 4, 5). For example, at the beginning of a sentence, abbreviation of the scientific name of the species, better not to be used (line 74).

2- I suggest to revise the title/caption of tables and figures, where the authors referring to the target species. It is suggested to use both common and scientific name of the species (as painted tree-rat (Callistomys pictus)) only in Figure 1 and Table 1. For the rest of figures and tables, only one (common or scientific) will be enough.

7. PLOS authors have the option to publish the peer review history of their article (what does this mean?). If published, this will include your full peer review and any attached files.

Reviewer #1: No

---

## [Editor Report · Acceptance letter]

10 Jan 2025

PONE-D-24-33359R1 

PLOS ONE

Dear Dr. Giné, 

I'm pleased to inform you that your manuscript has been deemed suitable for publication in PLOS ONE. Congratulations! Your manuscript is now being handed over to our production team.

Kind regards, 

on behalf of

Dr. Daniel de Paiva Silva 

Academic Editor

PLOS ONE